# Obesity-Induced MASLD Is Reversed by Capsaicin via Hepatic TRPV1 Activation

**DOI:** 10.3390/cimb47080618

**Published:** 2025-08-04

**Authors:** Padmamalini Baskaran, Ryan Christensen, Kimberley D. Bruce, Robert H. Eckel

**Affiliations:** 1Department of Pharmaceutical Sciences, College of Pharmacy, Howard University College of Pharmacy, Washington, DC 20059, USA; 2Molecular Signaling Laboratory, Department of Pharmaceutical Sciences, College of Pharmacy, University of Wyoming, Laramie, WY 82071, USA; ryan.christensen314@gmail.com; 3Anschutz Medical Campus, University of Colorado, Aurora, CO 80045, USA; kimberley.bruce@cuanschutz.edu (K.D.B.); robert.eckel@cuanschutz.edu (R.H.E.)

**Keywords:** MASLD, TRPV1, mitochondria, obesity, diabetes, insulin resistance, fatty acid oxidation

## Abstract

Background and Objectives: Metabolic dysfunction-associated steatotic liver disease (MASLD) is a progressive liver disorder associated with metabolic risk factors such as obesity, type 2 diabetes, and cardiovascular disease. If left untreated, the accumulation of excess hepatic fat can lead to inflammation, fibrosis, cirrhosis, hepatocellular carcinoma, and ultimately liver failure. Capsaicin (CAP), the primary pungent compound in chili peppers, has previously been shown to prevent weight gain in high-fat diet (HFD)-induced obesity models. In this study, we investigated the potential of dietary CAP to prevent HFD-induced MASLD. Methods: C57BL/6 mice were fed an HFD (60% kcal from fat) with or without 0.01% CAP supplementation for 26 weeks. We evaluated CAP’s effects on hepatic fat accumulation, inflammation, and mitochondrial function to determine its role in preventing MASLD. Results: CAP acts as a potent and selective agonist of the transient receptor potential vanilloid 1 (TRPV1) channel. We confirmed TRPV1 expression in the liver and demonstrated that CAP activates hepatic TRPV1, thereby preventing steatosis, improving insulin sensitivity, reducing inflammation, and enhancing fatty acid oxidation. These beneficial effects were observed in wild-type but not in TRPV1 knockout mice. Mechanistically, CAP-induced TRPV1 activation promotes calcium influx and activates AMPK, which leads to SIRT1-dependent upregulation of PPARα and PGC-1α, enhancing mitochondrial biogenesis and lipid metabolism. Conclusions: Our findings suggest that dietary CAP prevents MASLD through TRPV1 activation. TRPV1 signaling represents a promising therapeutic target for the prevention and management of MASLD in individuals with metabolic disorders.

## 1. Introduction

Metabolic dysfunction-associated steatotic liver disease (MASLD) is the most prevalent chronic liver condition in Western populations and is strongly linked to metabolic disorders such as obesity and type 2 diabetes [1]. MASLD is characterized by pathological accumulation of triglycerides in hepatocytes occurring in the absence of significant alcohol consumption, driven primarily by insulin resistance-induced hepatic lipogenesis [2]. This excessive lipid buildup initiates a pathogenic cascade involving oxidative stress, mitochondrial and endoplasmic reticulum dysfunction, and chronic inflammation [3]. If unresolved, MASLD can progress from simple steatosis to steatohepatitis, fibrosis, cirrhosis, and ultimately hepatocellular carcinoma [4], and is associated with increased risk of cardiovascular disease, chronic kidney disease, and all-cause mortality [5].

The C57BL/6 mouse, when fed a high-fat diet (HFD) or methionine- and choline-deficient diet, is a widely accepted preclinical model for MASLD, as it recapitulates key human features such as obesity, hyperinsulinemia, hepatic inflammation, and liver injury [6,7]. At the molecular level, AMP-activated protein kinase (AMPK) and Sirtuin-1 (SIRT1) are pivotal metabolic sensors that regulate hepatic energy homeostasis, fatty acid oxidation, and mitochondrial integrity [8,9,10]. These regulators converge on peroxisome proliferator-activated receptor γ coactivator 1α (PGC-1α), a master transcriptional co-regulator that modulates gluconeogenesis, lipid metabolism, and oxidative stress responses via PPARα and Uncoupling protein 2 (UCP2) [11,12,13].

The transient receptor potential vanilloid 1 (TRPV1) channel is a nonselective calcium-permeable cation channel known for its roles in thermogenesis [14,15] and nociception [16]. Capsaicin (CAP), the bioactive compound in chili peppers, is a potent TRPV1 agonist. Our previous work has shown that CAP activates TRPV1, elevates intracellular calcium, and stimulates CAMKK2-dependent AMPK phosphorylation, thereby promoting thermogenesis via transcriptional regulation of metabolic genes in adipose tissue [14,15].

Although CAP’s anti-obesity and lipid-lowering effects via TRPV1 activation have been demonstrated [17], its therapeutic potential against MASLD and associated insulin resistance remains poorly defined. In this study, we show that CAP suppresses MASLD development in HFD-fed obese mice via hepatic TRPV1-mediated activation of the AMPK-SIRT1 signaling axis. We further demonstrate that TRPV1 is expressed in liver tissue and its activation by CAP enhances PPARα, PGC-1α, and UCP2 expression, promoting fatty acid oxidation, reducing hepatic lipid burden, and inhibiting MASLD progression.

## 2. Materials and Methods

### 2.1. HFD-Induced MASLD Mouse Model

The mouse model of MASLD was developed by feeding mice an HFD for 26 weeks. The University of Wyoming Institutional Animal Care and Use Committee approved all animal care and experimental procedures. Animal studies are reported in compliance with the ARRIVE guidelines [18]. Adult wild-type (WT) and TRPV1^−/−^ (B6.129X1-Trpv1 tm1Jul/J) mice purchased from Jackson Laboratory, Bar Harbor, ME, USA, were housed in the research animal facility and maintained at 23 °C at the School of Pharmacy, University of Wyoming. Mice were bred according to the approved breeding protocol. Both sexes were used in this study. Mice were housed in groups of four, and proper husbandry care was exercised according to the recommendations of the Institutional Animal Care and Use Committee of the University of Wyoming. Mice were allowed access to the different diets and water ad libitum. HFD was obtained from Research Diets Inc., New Brunswick, NJ, USA.

### 2.2. Experimental Procedures

A total of 240 mice were used in the study, with 40 mice per group (20 males and 20 females) across six experimental groups: WT-NCD, WT-HFD, WT-HFD + CAP, TRPV1^−/−^-NCD, TRPV1^−/−^-HFD, and TRPV1^−/−^-HFD + CAP. Mice were housed in groups of four per cage, grouped by sex (either 4 males or 4 females). The sample size was determined based on prior pilot experiments and power calculations (α = 0.05, power = 80%) to detect significant differences in metabolic and histological outcomes between genotypes and dietary treatments. The in vivo and in vitro data in this study were obtained, and their analyses were carried out, without the knowledge of the treatment groups (blinded).

Capsaicin (≥95% purity; product number M2028, Millipore Sigma, St. Louis, MO, USA), derived from *Capsicum* species, was used for the intervention. CAP was incorporated into the HFD at a concentration of 0.01% using the geometric dilution method to ensure uniform mixing.

The 0.01% capsaicin concentration was selected based on published literature and our sub chronic safety and dose–response studies, in which we tested capsaicin concentrations ranging from 0.001% to 0.01%. In those studies, capsaicin was found to be effective at concentrations between 0.003% and 0.01% in reducing weight gain and improving metabolic parameters. Furthermore, in our previous molecular studies elucidating the mechanism of action underlying capsaicin’s weight loss effects, we used 0.01% capsaicin as a standard dose. To maintain consistency and relevance, we applied the same 0.01% concentration in this study to investigate capsaicin’s effect on MASLD.

All mice were fed a normal diet for the first six weeks. Starting from week 6, WT and TRPV1^−/−^ mice were randomly assigned into three groups and fed with the three different diets—normal chow diet (NCD), the high-fat diet (HFD; 60 kcal% fat; D-12492, Research Diets, New Brunswick, NJ, USA) or HFD with added capsaicin (0.01%; HFD + CAP). These diets were fed until week 32. At the end of the study, the mice were anesthetized (Ketamine 80 mg/Kg and Xylazine—10 mg/Kg by intraperitoneal injection), and the blood was collected by cardiac puncture in serum separation vacutainer tubes. The mice were then euthanized, and the liver was isolated and snap-frozen in liquid nitrogen and stored at −80 °C for further analysis.

Liver weight: The liver was carefully dissected, washed in ice-cold phosphate-buffered saline (PBS) and pat dried and weighed using a precision scale.

Liver triglyceride content: The liver triglyceride content was measured using a kit from Cayman Chemical (Triglyceride Calorimetric Assay kit # 10010303).

Liver histology sections—hematoxylin and eosin staining: A small amount of isolated liver tissues were fixed in 10% formalin solution, followed by embedding in paraffin. 8-micron paraffin sections were mounted onto glass slides and deparaffinized using xylene and ethanol solutions. The hydrated sections were stained with hematoxylin for nuclei and eosin for cytoplasm before dehydration and then cover-slipped using permount.

Liver histology sections—Oil Red O staining: Freshly harvested liver was embedded in a cryomold with enough optimal cutting temperature (OCT) compound and frozen using liquid nitrogen, and the frozen block in foil was stored at −80 °C. 7–10 microns of the tissue was sectioned, fixed in 10% formalin, followed by isopropanol and Oil Red O solution for the period mentioned in the protocol [19]. The slides were then washed in isopropanol and de-ionized water, stained the nucleus with hematoxylin, and cover-slipped with an aqueous mounting medium.

Liver total cholesterol content: The total cholesterol content in the liver was determined by the manufacturer’s protocol for the kit (MBS269999) from MyBioSource, San Diego, CA, USA.

Serum LDL cholesterol (LDL), HDL cholesterol (HDL) levels: Plasma samples (200 µL) were chromatographed via fast protein liquid chromatography (FPLC) using two Superose 6 columns in series [20] using size exclusion, the absorbance of the eluted samples was measured at 280 nm, allowing identification of each lipoprotein class via protein content. Fractions containing either LDL-C, or HDL-C were pooled, and cholesterol content was quantified using a commercially available kit (Cayman Chemical Company, Ann Arbor, MI, USA) following the manufacturer’s instructions.

Insulin tolerance test: The mice were fasted overnight. They were injected with a bolus of 0.5 U/Kg insulin (Novolog 100 IU/mL). The blood glucose levels were measured at 0, 30, 60, 90, 120, and 150 min after injection using a glucometer (Accuchek, Roche, Indianapolis, IN, USA). The blood glucose was monitored in a small amount of blood drawn after making a prick in the tip of the tail vein using a lancet. Homa insulin resistance was calculated using the formula Fasting insulin (µU/mL) × fasting glucose (mM)/22.5.

Quantitative RT-PCR measurement: Livers were collected from 6 cohorts of mice. The experiment was performed in triplicate for statistical analysis. Tri reagent (Sigma, St. Louis, MO, USA) was used for total RNA isolation. Quantitect reverse transcription kit (Qiagen, Valencia, CA, USA) was used for synthesizing cDNA from RNA. Quantitect SYBR green PCR kit was used for real-time PCR using a Q5plex PCR system. The reaction volume was 20 μL, and 18s mRNA was used as the reference gene. The de novo lipogenesis gene mRNA (Chrebp, SREBP-1c, SCD-1, CD-36, fas, and elovl-6) was determined. The primer sequence is tabulated in Table 1. Quantitative RT-PCR kits were obtained from Qiagen, USA. We also performed RT-PCR for the mitochondrial biogenesis gene PGC-1α and TRPV1.

Inflammatory Cytokines using ELISA: The frozen liver tissue was thawed on ice by adding 1 mL of PBS containing protease inhibitors. The tissues were homogenized using a tissue homogenizer and the lysate was centrifuged at 164× *g*, for 10 min in a refrigerated centrifuge and the cytokine levels were analyzed in the supernatant. The inflammatory cytokines were measured as per the manufacturer’s protocol using commercially available ELISA kits (Biovision, San Jose, CA, USA) in a Tecan plate reader.

Mitochondrial DNA Content: The mitochondrial DNA content was measured as mentioned before [21]. Frozen liver tissues were pulverized using a mortar and pestle and transferred into microcentrifuge tubes. The genomic DNA was isolated as per the manufacturer’s protocol (GDI-3, Sigma, St. Louis, MO, USA). The DNA concentration was determined using a Nanodrop. Real-time QPCR was performed in diluted DNA samples using 10 μM of the following primers: mtDNA (16s rRNA—stable part of mtDNA), forward 5′-CCGCAAGGGAAAGATGAAAGAGAC-3′, Reverse 5′-TCGTTTGGTTTCGGGGGTTTC-3′. The nuclear-specific PCR primers (hexokinase) are Forward-5′-GCCAGCCTCTCCTGATTTTAGTGT-3′, Reverse-5′-GGGAACACAAAAGACCTCTTCTGG-3′. The PCR program used for amplification was as follows: 20 min at 95 °C, followed by 50–55 cycles of 15 s at 95 °C, 20 s at 58 °C, and 20 s at 72 °C. The post-run melting curves were analyzed to verify amplification.

### 2.3. Immunoblotting

The immunoblotting protocol was performed as mentioned before [14,15]. The liver was washed with chilled PBS and homogenized in lysis buffer (50 mM Tris pH 7.5, 250 mM sodium chloride, 0.5% NP40, 0.5% sodium deoxycholate, 2 mM EDTA, 0.5 mM dithiothreitol, 1 mM sodium orthovanadate and protease inhibitor cocktail) and centrifuged at 20,817× *g* for 20 min to remove tissue debris. The protein concentration was determined using the Bradford method, and equal amounts of protein were separated by SDS-PAGE, transferred to a nitrocellulose membrane, and immunoblotted with specific antibodies. Sources of antibodies are shown in Table 2.

Fatty acid oxidation Assay (FAO) was performed as previously described [22]. Briefly, the liver was isolated from the mice and lysed in STE buffer (sucrose 0.25 M, Tris 10 mM, EDTA 1 mM, pH-7.4) using a Dounce homogenizer. The mitochondrial fraction was isolated by centrifuging at 720× *g* for 10 min at 4 °C. 40 µL of the mitochondrial fraction was added to oxidation reaction mix (sucrose 100 mM, Tris 10 mM, potassium dihydrogen phosphate 5 mM, EDTA 0.2 mM, KCl 80 mM, MgCl2 1 mM, L-carnitine—2 mM, L-malate-0.1 mM, coenzyme A 0.5 mM, ATP—2 mM, DTT—1 mM, Cold palmitate 500 µM/tube, Radioactive 1-^14^C-palmitate—0.4 µCi/tube, pH—7.4) and incubated at 37 °C for 60 min. The reaction stopped using 1 M perchloric acid. The long-chain fatty acids were precipitated by centrifugation at 14,000 rpm for 10 min, and radioactivity was measured using a beta scintillation counter. Specific activity was determined by dividing counts per minute (CPM) by total nanomoles of palmitate (cold + hot)/tube and expressed as nmol/mg/min.

### 2.4. Data and Statistical Analyses

Data are expressed as mean ± SEM. Student T test and ANOVA evaluated the statistical significance of population means. *p* < 0.05 was deemed to constitute the threshold for statistical significance for data comparison. Post hoc analyses were carried out only when the means were significant for *p* < 0.05, and there was no significant variance in homogeneity. Graphs from the analyzed data were plotted using Microcal Origin 6.0 software.

### 2.5. Materials

All chemicals were obtained from Sigma, St. Louis, MO, USA.

## 3. Results

### 3.1. Effect of CAP on Liver Weight, Triglyceride (TG) Content, and Histology

We previously demonstrated that both WT and TRPV1^−^/^−^ mice fed an HFD for 32 weeks exhibit significant weight gain [14]. Supplementation of the HFD with 0.01% CAP prevented weight gain in WT mice but had no effect in TRPV1^−^/^−^ mice, indicating a TRPV1-dependent response. Upon liver tissue isolation, substantial hepatic fat accumulation was observed in both HFD-fed WT and TRPV1^−^/^−^ mice. In contrast, CAP-fed WT mice showed no visible hepatic fat deposition and exhibited normal vascular architecture comparable to that of NCD-fed controls. However, TRPV1^−^/^−^ mice fed HFD ± CAP retained a steatotic appearance similar to HFD-only controls (Figure 1A).

Liver weight, normalized to tibial length, was significantly increased in HFD-fed mice, serving as a surrogate marker of hepatic steatosis (Figure 1B). Consistently, hepatic triglyceride (TG) levels were elevated in both WT and TRPV1^−^/^−^ mice on HFD, confirming steatosis. CAP feeding significantly reduced intrahepatic TG accumulation in WT mice but not in TRPV1^−^/^−^ mice, supporting a TRPV1-dependent mechanism (Figure 1C).

Histological analysis using hematoxylin and eosin (H&E) staining revealed large intracellular vacuoles indicative of steatosis in the livers of HFD-fed WT and TRPV1^−^/^−^ mice.

These vacuoles were larger and more prominent in TRPV1^−^/^−^ mice, irrespective of CAP supplementation. In contrast, CAP-fed WT livers displayed histological features comparable to NCD-fed controls (Figure 1D,E). Oil Red O staining further confirmed these findings, with intense lipid droplet accumulation in HFD-fed and TRPV1^−^/^−^ livers, while CAP-fed WT livers showed minimal lipid deposition (Figure 1F).

### 3.2. First Hit—Insulin Resistance

Insulin resistance is widely recognized as the primary initiating event or “first hit” in the pathogenesis of hepatic fat accumulation and MASLD [23]. Impaired insulin signaling promotes hepatic lipogenesis, contributing to triglyceride buildup and disease progression. In our study, HFD-fed WT mice displayed clear signs of insulin resistance, evidenced by impaired glucose clearance following insulin injection. In contrast, HFD + CAP-fed WT mice demonstrated improved insulin sensitivity, as reflected by a greater reduction in blood glucose levels post-insulin administration (Figure 2A).

However, TRPV1^−^/^−^ mice on HFD ± CAP exhibited only a modest decline in blood glucose levels after insulin injection, indicating persistent insulin resistance and a lack of CAP efficacy in the absence of TRPV1. The insulin tolerance test results were quantified using the area under the curve (AUC), shown in Figure 2B.

To further assess insulin sensitivity, we calculated the homeostatic model assessment of insulin resistance (HOMA-IR), where higher values indicate greater insulin resistance [9]. A HOMA-IR between 0.5 and 1.4 is considered normal; values above 1.9 indicate early insulin resistance, while values exceeding 2.9 reflect significant insulin resistance. HFD-fed WT mice and TRPV1^−^/^−^ mice (HFD ± CAP) exhibited elevated HOMA-IR scores, indicating marked insulin resistance. In contrast, HFD + CAP-fed WT mice and NCD-fed controls had HOMA-IR values near 0.5, suggesting preserved insulin sensitivity (Figure 2C).

These findings support that CAP improves insulin sensitivity in a TRPV1-dependent manner, blocking the initial “first hit” in MASLD development.

### 3.3. Second Hit—Oxidative Stress and Inflammation

Following hepatic fat accumulation, the “second hit” in the progression of MASLD involves oxidative (redox) stress and inflammation, which drive the transition from simple steatosis to steatohepatitis and ultimately fibrosis. Key pro-inflammatory cytokines, including tumor necrosis factor-alpha (TNF-α), Interleukin-1 beta (IL-1β), interleukin-6 (IL-6), and C-reactive protein (CRP), play central roles in promoting hepatic injury and oxidative stress.

To assess this inflammatory response, we measured the hepatic levels of these cytokines in six experimental cohorts. Livers from HFD-fed WT mice exhibited a significant increase in inflammatory markers, consistent with a heightened redox and inflammatory state. In contrast, NCD-fed WT mice and HFD + CAP-fed WT mice displayed markedly lower levels of these cytokines, indicating an attenuation of liver inflammation by CAP.

Strikingly, TRPV1^−^/^−^ mice fed HFD, with or without CAP supplementation, showed the highest levels of hepatic inflammatory cytokines among all groups, underscoring the critical role of TRPV1 in mediating CAP’s anti-inflammatory effects (Figure 3). These findings demonstrate that CAP mitigates HFD-induced hepatic inflammation through TRPV1 activation, thereby interrupting the second hit in MASLD pathogenesis and potentially preventing disease progression.

### 3.4. Lipoprotein Profile in the Mouse Fed with NCD, HFD ± CAP

MASLD is frequently associated with dyslipidemia, characterized by elevated total cholesterol and low-density lipoprotein cholesterol (LDL-C), along with reduced high-density lipoprotein cholesterol (HDL-C). In our study, WT mice fed an NCD or an HFD + CAP exhibited a favorable plasma lipid profile, with lower levels of total cholesterol and LDL-C and elevated HDL-C.

In contrast, WT mice fed an HFD alone, as well as TRPV1^−^/^−^ mice fed HFD ± CAP, displayed an abnormal lipoprotein profile. These groups showed significantly increased total cholesterol and LDL-C levels, along with decreased HDL-C concentrations, indicative of dyslipidemia (Figure 4). This lipid imbalance is closely linked to insulin resistance and may further contribute to the development and progression of MASLD.

These findings highlight the role of TRPV1 in regulating lipid metabolism and suggest that dietary CAP improves dyslipidemia in a TRPV1-dependent manner, potentially mitigating one of the key metabolic risk factors for MASLD.

### 3.5. De Novo Lipogenesis (DNL) in the Liver

DNL is significantly upregulated under insulin-resistant conditions and plays a key role in hepatic triglyceride accumulation, thereby promoting the development of MASLD. In our study, the expression of key DNL-associated genes, including ChREBP, SREBP-1c, CD36, FAS, SCD-1, and Elovl-6, was markedly elevated in the livers of WT mice fed an HFD, as well as in TRPV1^−^/^−^ mice fed HFD with or without CAP.

In contrast, CAP supplementation significantly suppressed the expression of these lipogenic genes in HFD-fed WT mice, correlating with reduced hepatic fat accumulation and protection against MASLD progression (Figure 5). Notably, CAP had no inhibitory effect on DNL gene expression in TRPV1^−^/^−^ mice, reinforcing the requirement for TRPV1 signaling in mediating CAP’s metabolic benefits.

These findings suggest that CAP prevents hepatic steatosis by inhibiting insulin resistance-driven DNL in a TRPV1-dependent manner, thereby disrupting a central pathogenic mechanism in MASLD.

### 3.6. TRPV1 Expression in the Liver

The data showed that HFD feeding induced hepatic steatosis in both WT and TRPV1^−^/^−^ mice. While CAP treatment failed to prevent fat accumulation in TRPV1^−^/^−^ mice, it effectively protected WT mice from steatosis. As a selective agonist of TRPV1, CAP’s protective effect appears to be mediated through TRPV1 activation. To determine whether TRPV1 is expressed in the liver, we performed immunoprecipitation using liver lysates. As shown in Figure 6, TRPV1 expression was detected in WT mice fed with NCD and HFD + CAP. However, TRPV1 was absent in the liver lysates of WT mice fed HFD alone and in TRPV1^−^/^−^ mice.

### 3.7. Energy Sensors AMPK and SIRT-1 in MASLD

It is well established that AMPK activation prevents lipid accumulation in the liver of HFD-fed mice by inhibiting lipogenesis and promoting fatty acid oxidation [9]. We previously demonstrated that CAP, through TRPV1-mediated Ca^2+^ influx, activates AMPK via phosphorylation at Thr172 [15]. To assess this pathway in the liver, we analyzed liver lysates and found that the ratio of phospho-AMPK (Thr172) to total AMPK was elevated in WT mice fed NCD and in those treated with CAP under HFD conditions. As shown in Figure 7, AMPK phosphorylation was markedly reduced in WT-HFD and TRPV1^−^/^−^ mice under both HFD and HFD + CAP conditions.

AMPK and SIRT-1 are known to function in coordination; AMPK activation enhances SIRT-1 activity, which in turn regulates downstream targets such as PGC-1α and PPAR-α-key mediators of mitochondrial function and fatty acid oxidation [24]. Consistent with this, we observed downregulation of SIRT-1 in liver lysates exhibiting reduced AMPK phosphorylation. The immunoblots are presented in Figure 8. Densitometric analysis showed higher SIRT-1 expression in liver samples from WT mice fed NCD and from WT mice treated with CAP under HFD conditions.

### 3.8. Regulators of Lipid Metabolism

In the liver, PPARα (peroxisome proliferator-activated receptor alpha) and PGC-1α work in concert to promote fatty acid oxidation and mitochondrial biogenesis [25]. PPARα functions as the primary transcription factor, while PGC-1α serves as its coactivator. Dysregulation of the PPARα/PGC-1α axis is associated with impaired fatty acid oxidation and increased lipid accumulation, as observed in MASLD [26]. CAP treatment upregulated both PPARα and PGC-1α protein expression in the livers of HFD-fed WT mice (Figure 8 and Figure 9). In contrast, TRPV1^−^/^−^ mice exhibited markedly reduced levels of both PPARα and PGC-1α, consistent with the severe hepatic steatosis observed in histological analyses (H&E and Oil Red O staining).

Moreover, mitochondrial DNA (mtDNA) content was significantly elevated in the livers of HFD + CAP-treated WT mice (Figure 10). Notably, TRPV1^−^/^−^ mice fed NCD had lower mtDNA content compared to WT-NCD controls, and mtDNA levels were further reduced in TRPV1^−^/^−^ mice under HFD ± CAP conditions relative to TRPV1^−^/^−^ NCD controls. These findings were corroborated by similar trends in the expression of the mitochondrial biogenesis marker, PGC-1α mRNA (Figure 10B).

### 3.9. Energy Dissipator, UCP-2

UCP-2 (uncoupling protein 2) is a mitochondrial protein expressed in the liver that plays a key role in lipid metabolism by enhancing fatty acid oxidation and reducing oxidative stress caused by lipid accumulation [27]. In our study, the increase in mitochondrial copy number observed in CAP-treated WT mice under HFD conditions was accompanied by elevated UCP-2 expression. In contrast, TRPV1^−^/^−^ mice fed HFD ± CAP exhibited markedly lower UCP-2 levels, as shown in Figure 9.

## 4. Discussion

MASLD is a common comorbidity associated with diet-induced obesity, contributing to a growing global health burden. While dietary and lifestyle interventions are effective preventative measures, sustained adherence remains a major challenge. Furthermore, resmetirom (also known as Rezdiffra), is the sole pharmacological agent currently approved for MASLD [28,29] highlighting an urgent unmet clinical need. Thus, identifying effective and mechanistically validated therapeutic strategies to combat MASLD is of paramount importance.

In this study, we demonstrate that oral supplementation with CAP, a bioactive component of chili peppers, significantly attenuates the progression of MASLD in a murine model of HFD-induced obesity. DNL genes such as ChREBP and SREBP-1c, transcription factors that upregulate key lipogenic enzymes, promoting fat synthesis in the liver. Fatty acid uptake into hepatocytes occurs through CD36, while FAS, SCD-1, and ELOVL-6 support triglyceride formation. Together, these genes drive hepatic fat accumulation.

In MASLD, inflammatory cytokines such as TNF-α, IL-6, IL-1β, and CRP play key roles in driving liver inflammation. TNF-α promotes hepatocyte injury, insulin resistance, and fibrogenesis. IL-6 contributes to chronic inflammation and worsens lipid metabolism dysregulation. IL-1β induces hepatocyte apoptosis and further amplifies inflammatory signaling. CRP, produced in response to IL-6, serves as a marker of systemic inflammation and correlates with the severity of MASLD. Persistent elevation of these cytokines leads to progression from simple steatosis to steatohepatitis (MASH), promoting fibrosis and liver dysfunction.

CAP-fed wild-type (WT) mice showed reduced hepatic steatosis (Figure 1), improved lipid profiles (Figure 2), enhanced insulin sensitivity (Figure 3), and suppressed hepatic inflammation (Figure 5). These beneficial effects were absent in TRPV1 knockout (TRPV1^−^/^−^) mice, confirming that CAP’s hepatoprotective actions are dependent on TRPV1 signaling, consistent with prior studies implicating TRPV1 as a metabolic regulator [14,15,30].

At the mechanistic level, CAP restored hepatic TRPV1 expression (Figure 6) and activated intracellular pathways known to modulate lipid metabolism. TRPV1 activation led to phosphorylation of AMPK at Thr172 and subsequent enhancement of SIRT-1 activity (Figure 7 and Figure 8), a pathway central to mitochondrial biogenesis and lipid oxidation [31,32,33]. This signaling cascade restored hepatic expression of PPARα, SIRT-1, and PGC-1α (Figure 8 and Figure 9), resulting in suppression of de novo lipogenesis (DNL) and promotion of fatty acid oxidation (FAO). These findings align with prior evidence linking AMPK and SIRT-1 to improved hepatic lipid metabolism and mitochondrial function in MASLD [33].

Functional validation was obtained through FAO assays, with CAP-fed WT mice exhibiting significantly increased [^14^C]-palmitate oxidation (160 ± 7.3 nmol/mg/min), compared to NCD-fed controls (110 ± 5.7 nmol/mg/min). In contrast, TRPV1^−^/^−^ mice exhibited markedly reduced FAO capacity under both NCD and HFD + CAP conditions (70 ± 3 and 73 ± 3 nmol/mg/min, respectively). Consistent with prior work, we observed CAP-induced upregulation of mitochondrial uncoupling protein-2 (UCP-2) [34,35], a key determinant of hepatic oxidative capacity. In line with prior studies, CAP has also been implicated in maintaining mitochondrial quality control [36], and HFD-fed mice displayed impaired mitophagy and mitochondrial dysfunction [37]. These results open new avenues to explore CAP’s involvement in mitochondrial biogenesis and mitophagy as potential mechanisms underlying its hepatoprotective effects.

Our initial investigation was specifically focused on understanding the molecular mechanism by which CAP inhibits weight gain. Upon tissue isolation at the end of the study, we observed that mice fed an HFD developed fatty liver, while this effect was notably inhibited in CAP-fed mice. These findings led us to further explore CAP’s role in attenuating MASLD in the HFD model. We plan to extend our studies to include fructose-fed juvenile rodent models, as high fructose consumption, commonly from sugary beverages and processed foods, is increasingly recognized as a major contributor to the rising incidence of MASLD in children.

CAP is well-established to promote systemic energy expenditure and reduce adiposity by inducing thermogenesis via TRPV1 activation in brown and beige adipose tissues [14,15]. These anti-obesity effects have been shown to indirectly improve liver health by reducing lipid influx and inflammation [38]. However, while our findings establish CAP as a TRPV1-dependent hepatoprotective agent in diet-induced obesity, it remains unclear whether these effects are mediated solely through systemic metabolic improvements or whether CAP also acts directly on the liver, independent of weight reduction. To address this critical question, future studies will utilize the methionine–choline-deficient (MCD) diet model, which induces steatosis, inflammation, and fibrosis without concurrent obesity or insulin resistance [39]. This model provides a unique opportunity to evaluate hepatic-specific mechanisms of CAP action in the absence of confounding metabolic factors. Furthermore, we plan to generate and study liver-specific TRPV1 knockout (TRPV1^LKO^) mice to determine whether hepatic TRPV1 expression is required for the observed metabolic benefits. Liver-specific knockout models have proven to be essential tools in elucidating organ-specific roles of lipid metabolism [40,41]. Together, these approaches will help clarify whether CAP’s effects on MASLD stem from direct hepatic TRPV1 activation or result primarily from its systemic anti-obesity properties. Importantly, MASLD can also occur in lean individuals, particularly in Asian populations and those with lipodystrophic syndromes, challenging the notion that obesity is a prerequisite for disease onset [42,43]. Therefore, assessing CAP and TRPV1 efficacy in non-obese MASLD models will be critical to determine the full translational scope of this therapeutic approach.

One notable limitation of CAP as an oral therapeutic is its low systemic bioavailability and inherent pungency, both of which may impact patient compliance. Despite these challenges, CAP administered ad libitum in the diet was sufficient to produce robust anti-MASLD effects in this study. Importantly, food intake was comparable between the HFD and HFD + CAP groups, as previously reported [44] indicating that the observed metabolic benefits were not attributable to reduced caloric intake but rather to CAP’s pharmacological action. To overcome these limitations and improve translational relevance, future studies will explore encapsulated, sustained-release and gastro-retentive CAP formulations, which are designed to prolong gastrointestinal residence time and enhance hepatic bioavailability. A recent pharmacokinetic analysis demonstrated that encapsulated formulations could achieve extended gastrointestinal retention and improved liver targeting [45]. Furthermore, our study demonstrates that pungent capsaicin (CAP) binds to and activates its highly selective receptor, TRPV1, with significantly greater efficacy than the non-pungent analog capsiate, as previously reported [46]. This highlights the importance of using pungent CAP for optimal TRPV1 activation. To enhance tolerability and clinical applicability, encapsulated formulations of pungent CAP offer a promising strategy to preserve its therapeutic efficacy while minimizing direct sensory exposure in the oral cavity, thereby improving patient compliance and translational potential.

CAP may also confer hepatoprotection via other mechanisms. It has been shown to activate nuclear factor erythroid 2-related factor 2 (Nrf2) [47,48], enhancing cellular antioxidant responses and reducing hepatic oxidative stress, a major contributor to MASLD progression. Additionally, CAP modifies the gut microbiota and reduces endotoxemia, implicating the gut–liver axis in its protective mechanism [49,50]. Recent studies also suggest crosstalk between TRPV1 and bile acid receptors (e.g., FXR, TGR5), GLP-1, and insulin signaling-all critical pathways in hepatic metabolic regulation [51,52,53].

In conclusion, our study highlights the therapeutic potential of CAP as a TRPV1-dependent modulator of hepatic lipid metabolism and mitochondrial function in MASLD. While current treatments like resmetirom address some aspects of the disease, the multifaceted effects of CAP—through the TRPV1–AMPK–SIRT-1–UCP-2 axis and possibly other pathways—offer a promising avenue for more comprehensive management. Future research focusing on liver-specific mechanisms, improved formulations, and efficacy in non-obese MASLD models will be critical to fully realize CAP’s clinical potential in addressing this growing global health challenge.

## Figures and Tables

**Figure 1 cimb-47-00618-f001:**
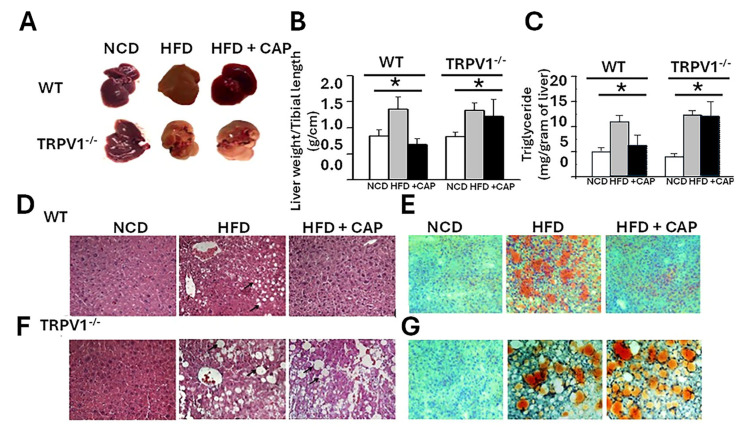
CAP suppresses hepatic steatosis. (**A**) Representative livers isolated from WT and TRPV1^−/−^ mice fed an NCD or HFD (±CAP) for 26 weeks. (**B**,**C**) Liver weight of mouse relative to tibial length and hepatic triglyceride content. (**D**,**E**) H&E-stained liver sections showing hepatic steatosis (arrows in (**D**,**F**)), and (**E**,**G**)—Oil Red O-stained liver sections indicating fat accumulation. Data are presented as mean ± SEM, *n* = 8 per group. * *p* < 0.05 by one-way ANOVA followed by post hoc test. NCD, normal chow diet; HFD, high-fat diet; CAP, capsaicin.

**Figure 2 cimb-47-00618-f002:**
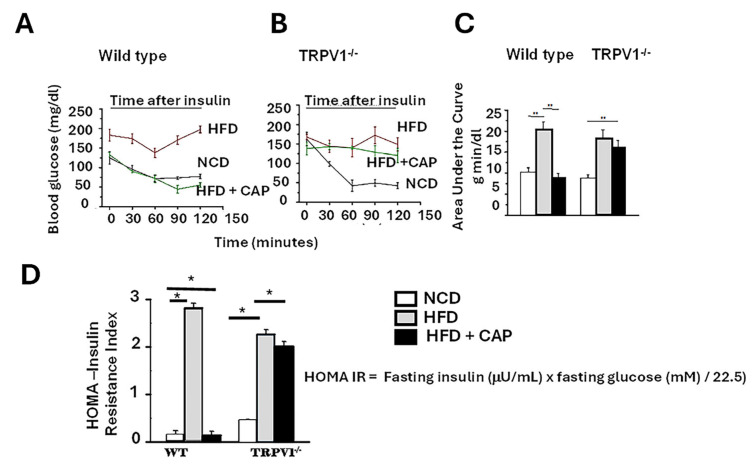
Dietary CAP inhibits insulin resistance in HFD model of obesity. HFD induces insulin resistance, which is counteracted by CAP only in WT mice. (**A**,**B**)—Fasting blood glucose levels in WT and TRPV1^-/-^ mice fed an NCD or HFD (±CAP) for 26 weeks, measured following administration of 1 U/kg insulin. (**C**) Area under the curve for panels (**A**,**B**). (**D**) Insulin resistance index (HOMA-IR, homeostatic model assessment) in the same groups. Data are presented as mean ± SEM, *n* = 8 per group. * *p* < 0.05, ** *p* < 0.01 by one-way ANOVA followed by post hoc test.

**Figure 3 cimb-47-00618-f003:**
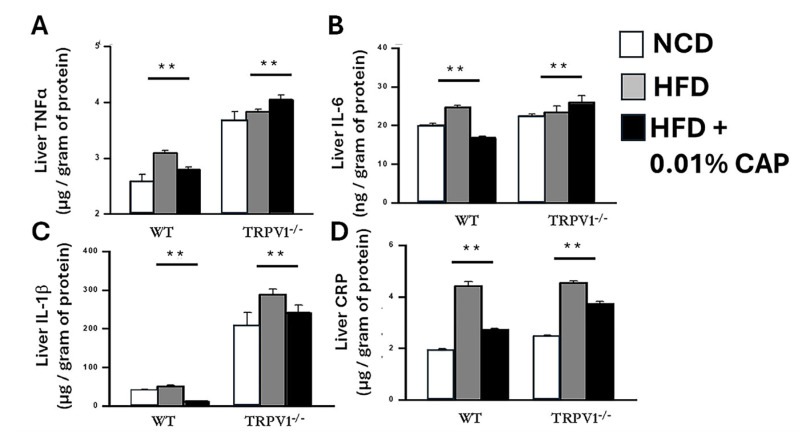
CAP reduces hepatic inflammatory cytokines. Levels of TNFα (**A**), IL-6 (**B**), IL-1β (**C**), and C-reactive protein (CRP; **D**) in liver lysates from WT and TRPV1^−/−^ mice. Data are presented as mean ± SEM, *n* = 8 per group. ** *p* < 0.01 by one-way ANOVA.

**Figure 4 cimb-47-00618-f004:**
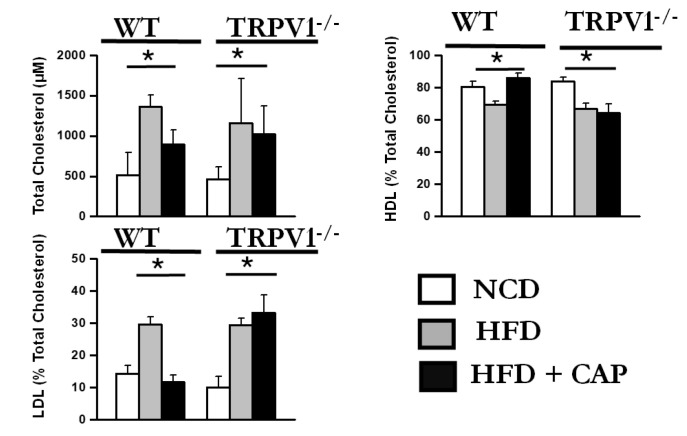
Dietary CAP regulates plasma cholesterol levels. Total cholesterol (TC), high-density lipoprotein cholesterol (HDL), low-density lipoprotein cholesterol (LDL) levels in the plasma of mice. Data are presented as mean ± SEM, *n* = 8 per group. * *p* < 0.05 by one-way ANOVA followed by post hoc test.

**Figure 5 cimb-47-00618-f005:**
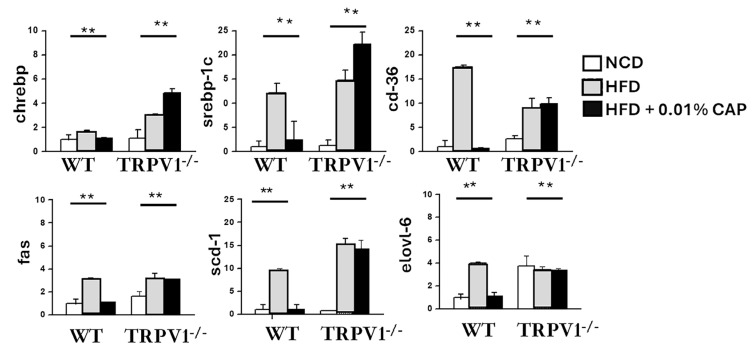
*TRPV1* deficiency increases expression of de novo lipogenesis (DNL) genes in mouse liver. Gene expression levels of key DNL pathway components were measured in liver tissues using quantitative RT-PCR. Data are presented as mean ± SEM, *n* = 8 per group. ** *p* < 0.01 by one-way ANOVA.

**Figure 6 cimb-47-00618-f006:**
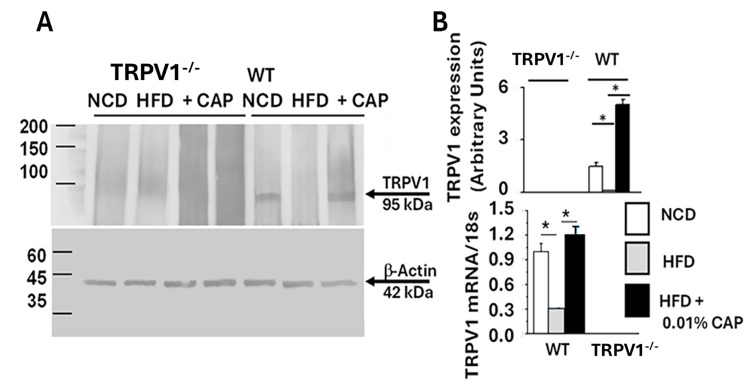
(**A**) Representative Western blot showing TRPV1 expression in liver lysates from WT and TRPV1^−/−^ mice fed an NCD or HFD (±CAP) for 26 weeks (top panel). β-Actin was used as a loading control in the precleared lysate (Bottom panel). (**B**) Quantification of TRPV1 band intensity normalized to β-actin (Upper panel), TRPV1 mRNA expression normalized to 18s RNA in liver tissues (lower panel). Data are presented as mean ± SEM, *n* = 3 independent liver samples from each cohort. * *p* < 0.05 by one-way ANOVA followed by post hoc test.

**Figure 7 cimb-47-00618-f007:**
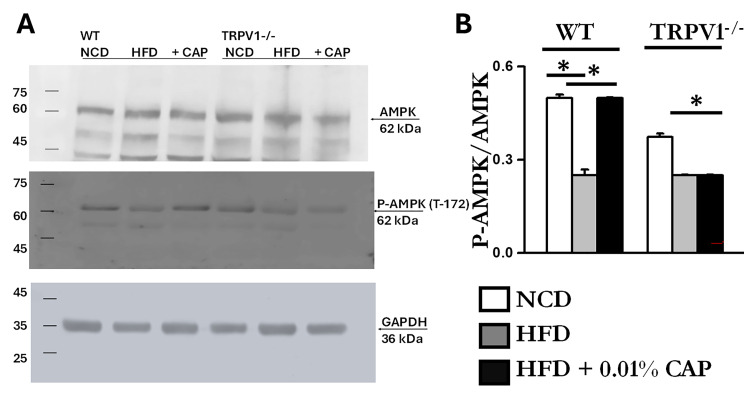
HFD suppresses AMPK phosphorylation, which is restored by CAP. (**A**) Representative immunoblots showing AMPK (top panel), phosphorylated AMPK (middle panel), and GAPDH (bottom panel) in the liver lysates from WT and TRPV1^−/−^ mice fed an NCD or HFD (±CAP) for 26 weeks (*n* = 6 experiments). (**B**) Quantification of the phosphor-AMPK to total AMPK ratio. Data are presented as mean ± SEM, *n* = 6 per group. * *p* < 0.05 by one-way ANOVA followed by post hoc test.

**Figure 8 cimb-47-00618-f008:**
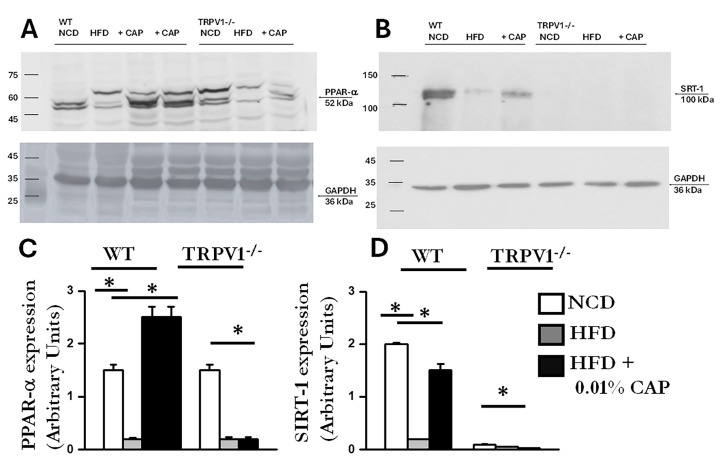
Expression of PPARα and SIRT-1 in liver lysates. (**A**,**B**) Representative immunoblots showing expression of PPARα and SIRT-1 in 40 μ g of liver lysates of WT and TRPV1^−/−^ fed an NCD or HFD (±CAP) diet for 26 weeks. (*n* = 6 experiments). (**C**,**D**) Quantification of band intensities, expressed as the ratio of PPARα to GAPDH and SIRT-1 to GAPDH, respectively. Data are presented as mean ± SEM, *n* = 6 per group. * *p* < 0.05 by one-way ANOVA followed by post hoc test; significantly different as indicated.

**Figure 9 cimb-47-00618-f009:**
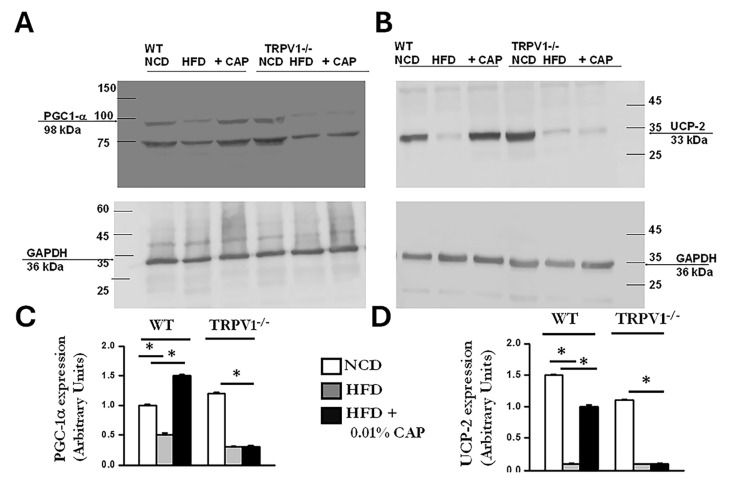
CAP restores PGC-1α and UCP-2 expression suppressed by HFD in WT but not TRPV1^−^/^−^ mice. (**A**,**B**) Representative immunoblots showing expression of PGC-1α and UCP-2 in liver lysates from WT and TRPV1^−^/^−^ mice fed an NCD or HFD (±CAP) for 26 weeks (*n* = 6 experiments). (**C**,**D**) Quantification of band intensities, expressed as the ratio of PGC-1α to GAPDH and UCP-2 to GAPDH, respectively. Data are presented as mean ± SEM, *n* = 6 per group. * *p* < 0.05 by one-way ANOVA followed by post hoc test; significantly different as indicated.

**Figure 10 cimb-47-00618-f010:**
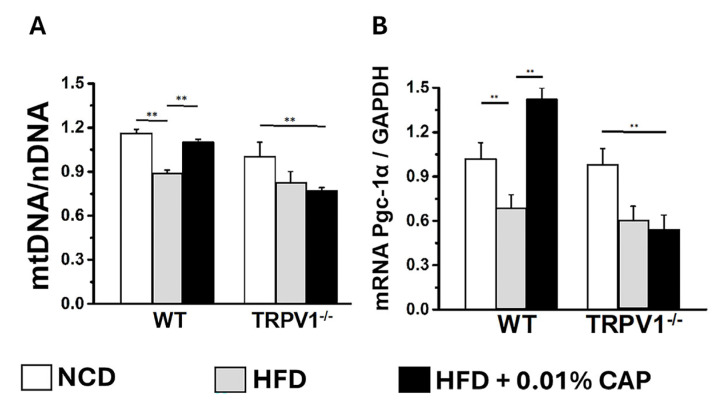
Mitochondrial DNA and PGC-1α levels in liver tissue: (**A**) Ratio of mitochondrial DNA (mtDNA) to nuclear DNA (nDNA) in liver samples from WT and TRPV1^−^/^−^ mice fed an NCD or HFD ± CAP for 26 weeks. Data are presented as mean ± SEM, *n* = 5 per group. ** *p* < 0.01 by one-way ANOVA. (**B**) PGC-1α mRNA expression in the same groups, showing that HFD suppresses PGC-1α expression, which is restored by CAP only in WT mice. Data are presented as mean ± SEM, *n* = 6 per group. ** *p* < 0.01 by one-way ANOVA.

**Table 1 cimb-47-00618-t001:** Primers used for RT-PCR. Primers were designed and then purchased from Integrated DNA Technologies (USA).

Gene (Accession Number)	Forward	Reverse
18s (X00686)	ACC GCA GCT AGG AAT AAT GGA	GCC TCA GTT CCG AAA ACC A
SCD1(NM_009127.4)	CTACAAGCCTGGCCTCCTGC	GGACCCCAGGGAAACCAGGA
ChREBP-α (NM_032951.3)	CGACACTCACCCACCTCTTC	TTGTTCAGCCGGATCTTGTC
Fas (NM_007988.3)	CCAGGAGAGACTGTCACAAGG	ACTGGGATCCTCTGAACACCT
Elovl6 (NM_130450.2)	TGCAGCATGACAACGACCAGTG	AATGGCAGAAGAGCACAAGGTAGC
SREBP-1c (BC056922.1)	ATCGGCGCGGAAGCTGTCGGGGTAGCGTC	ACTGTCTTGGTTGTTGATGAGCTGGAGCAT
CD-36 (NM_001159557.2)	GATGACGTGGCAAAGAACAG	TCCTCGGGGTCCTGAGTTAT
PGC-1α (NR_17532)	AGAGAGGCAGAAGCAGAAAGCAAT	ATTCTGTCCGCGTTGTGTCAGG
GAPDH (NM_001411844)	CGTGCCGCCTGGAGAAACC	TGGAAGAGTGGGAGTTGCTGTTG
TRPV1 (NM_001001445)	CAACAAGAAGGGGCTTACACC	TCTGGAGAATGTAGGCCAAGAC

**Table 2 cimb-47-00618-t002:** List of antibodies.

Antibodies	Dilution Used	Source	Catalog No.
TRPV1	1:100	Santa Cruz Biotechnology, Inc., Dallas, TX, USA	SC-28759
AMPKα	1:1000	Cell Signaling Inc., Danvers, MA, USA	2532s
Phospho-AMPKα	1:500	Cell Signaling Inc., Danvers, MA, USA	2535s
GAPDH	1:1000	Cell Signaling Inc., Danvers, MA, USA	2118s
UCP2	1:1000	Cell Signaling Inc., Danvers, MA, USA	89326
SIRT1	1:1000	Cell Signaling Inc., Danvers, MA, USA	2028s
β-actin	1:1000	Cell Signaling Inc., Danvers, MA, USA	4970
PPARα	1:500	Novus Biologicals, Danvers, MA, USA	NB600 - 636
PGC-1α	1:1000	Abcam Limited, Waltham, MA, USA	Ab191838

## Data Availability

The raw data supporting the conclusions of this article are available from the authors upon reasonable request.

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
