# Peer review of "Obesity-Induced MASLD Is Reversed by Capsaicin via Hepatic TRPV1 Activation"

_cimb, 2025, doi:10.3390/cimb47080618_

Round 1

Reviewer 1 Report

Comments and Suggestions for Authors

The authors studied the effects of Capsaicin on MASLD-associated hepatic steatosis, insulin resistance, oxidative stress, and inflammation. Besides that, they investigated the molecular mechanisms involved in capsaicin's beneficial effects, studying de novo lipogenesis, the energy sensors AMPK and SIRT-1, and regulators of liver metabolism. I congratulate the authors for including males and females in the study. I recommend minor revisions to improve the quality of the presentation. My comments are as follows:

  1. Materials and Methods (Page 3, Line 94): I recommend specifying the anesthetic and its dose.
  2. Figures: Please standardize the bar graphs in terms of appearance and group positions.
  3. Legends: I recommend standardizing the Legends, including mean+-SEM or mean+-SD, n=?, p-value, and statistical test.
  4. Figure 1 (Page 6): Despite the statistical difference described, there is no symbol of significance in graphs B and C.

Author Response

Reviewer:

The authors studied the effects of Capsaicin on MASLD-associated hepatic steatosis, insulin resistance, oxidative stress, and inflammation. Besides that, they investigated the molecular mechanisms involved in capsaicin's beneficial effects, studying de novo lipogenesis, the energy sensors AMPK and SIRT-1, and regulators of liver metabolism. I congratulate the authors for including males and females in the study. I recommend minor revisions to improve the quality of the presentation. My comments are as follows:

Comment 1: Materials and Methods (Page 3, Line 94): I recommend specifying the anesthetic and its dose.

Response: Thank you for your suggestion. We have revised the Materials and Methods section to specify the anesthetic and its dosage as follows:. “At the end of the study, the mice were anesthetized (Ketamine 80 mg/Kg and Xylazine – 10 mg/Kg by intraperitoneal injection), and the blood was collected by cardiac puncture in serum separation vacutainer tubes. “

Comment 2: Figures: Please standardize the bar graphs in terms of appearance and group positions.

Response: Thank you for your comment. We have carefully reviewed and standardized all bar graphs in the manuscript to ensure consistency in appearance, including uniform group positions, color schemes, and labeling across all figures.

Comment 3: Legends: I recommend standardizing the Legends, including mean+-SEM or mean+-SD, n=?, p-value, and statistical test.

Response: Thank you for your helpful suggestion. We have reviewed and revised all figure legends to ensure consistency and clarity. Each legend now includes standardized information specifying whether data are presented as mean ± SEM or mean ± SD, the sample size (n), p-values, and the statistical test used. These updates have been incorporated into the revised manuscript.

Comment 4: Figure 1 (Page 6): Despite the statistical difference described, there is no symbol of significance in graphs B and C.

Response: Thank you for bringing this to our attention. We have reviewed Figure 1 and identified the omission of significance symbols in panels B and C. The appropriate significance markers have now been added to reflect the statistical differences described in the results. The corrected figure is included in the revised manuscript.

Reviewer 2 Report

Comments and Suggestions for Authors

This original article by Padmamalini Baskaran et al., is an excellent work that borrows different promising objectives in the treatment of MASLD, from molecular, transcriptional, and post-transcriptional components, to the origin of the treatment proposed as CAP. The work is well-structured and well-conducted. I enjoyed reviewing this manuscript, however this reviewer has the following suggestions:

  • Line 56: It is suggested to define UPC2 before its abbreviation. The same for all abbreviations in the text.
  • In the Materials and Methods section, the sample size for each of the six study groups is not clear. Were there 40 mice for each group? The authors mention that mice were divided into boxes of 4 females and 4 males. It is suggested to clarify this paragraph and briefly explain how the authors calculated the sample size.
  • It is suggested to describe how the CAP was obtained for the intervention, species, processing, and vehicle, as well as how the treatment dose was defined. In some sections, the authors mention that they have previous work with CAP; however, it is important that this information also be declared in the methodology of this article.
  • Concerning the previous comment, the dose of CAP used in this work, how much does it represent what a human consumes in the average diet?
  • Figures 3 and 5 are missing in the results.
  • It is suggested to expand the discussion regarding the genes and proinflammatory cytokines evaluated.
  • MASLD is also associated with high fructose diets, which also promote increased lipogenesis, dyslipidemia, and insulin resistance, so it would be interesting to discuss further the choice of diet for this study.

Author Response

Reviewer 1:

This original article by Padmamalini Baskaran et al., is an excellent work that borrows different promising objectives in the treatment of MASLD, from molecular, transcriptional, and post-transcriptional components, to the origin of the treatment proposed as CAP. The work is well-structured and well-conducted. I enjoyed reviewing this manuscript, however this reviewer has the following suggestions:

Comment 1: Line 56: It is suggested to define UPC2 before its abbreviation. The same for all abbreviations in the text.

Response: Thank you for your suggestion. I have expanded the abbreviation for UCP2 at its first mention in the revised version of the manuscript and highlighted it accordingly. I have also reviewed the entire manuscript to ensure that all abbreviations are defined the first time they appear.

Comment 2: In the Materials and Methods section, the sample size for each of the six study groups is not clear. Were there 40 mice for each group? The authors mention that mice were divided into boxes of 4 females and 4 males. It is suggested to clarify this paragraph and briefly explain how the authors calculated the sample size.

Response: Thank you for your valuable comment. We have clarified the sample size and housing details in the Materials and Methods section. Specifically, a total of 240 mice were used in the study, with 40 mice per group (20 males and 20 females) distributed across six experimental groups: WT-NCD, WT-HFD, WT-HFD+CAP, TRPV1−/−-NCD, TRPV1−/−-HFD, and TRPV1−/−-HFD+CAP.

Mice were housed in groups of four per cage, separated by sex (either 4 males or 4 females per cage). The sample size was determined based on prior pilot experiments and power calculations (α = 0.05, power = 80%) designed to detect significant differences in metabolic and histological outcomes between genotypes and dietary treatments.

This clarification has been added to the revised manuscript for better transparency. The edits are highlighted in the revised version.

Comment 3: It is suggested to describe how the CAP was obtained for the intervention, species, processing, and vehicle, as well as how the treatment dose was defined. In some sections, the authors mention that they have previous work with CAP; however, it is important that this information also be declared in the methodology of this article.

Response: Thank you for your insightful suggestion. We have now included a detailed description of the capsaicin (CAP) intervention in the Materials and Methods section. Capsaicin (≥95% purity; product number M2028, Millipore Sigma, USA), derived from Capsicum species, was used for the intervention. It was incorporated into the high-fat diet (HFD) at a concentration of 0.01% using the geometric dilution method to ensure uniform mixing.

The 0.01% capsaicin concentration was selected based on published literature and our own sub-chronic safety and dose-response studies, where concentrations ranging from 0.001% to 0.01% were tested (ref 44). Capsaicin was effective at concentrations between 0.003% and 0.01% in reducing weight gain and improving metabolic parameters. Furthermore, in our previous molecular studies elucidating the mechanism of action underlying capsaicin’s weight loss effects, we consistently used 0.01% as the standard dose. To maintain consistency and relevance, the same concentration was applied in this study to investigate capsaicin’s effect on metabolic dysfunction-associated steatotic liver disease (MASLD).

This information has been incorporated into the revised manuscript to provide a clearer methodological context.

Comment 4: Concerning the previous comment, the dose of CAP used in this work, how much does it represent what a human consumes in the average diet?

Response: We appreciate the reviewer’s important question regarding the translational relevance of the CAP dose used in our study.

The CAP used in this study was purified capsaicin (≥95% purity; product number M2028, Millipore Sigma, USA), corresponding to approximately 16 million Scoville Heat Units (SHU). This is considerably higher in concentration than the CAP content typically found in foods such as jalapeño peppers or other commonly consumed chili varieties. In regular dietary habits, even among individuals with high spice consumption or those taking capsaicin-containing supplements, the amount of pure capsaicin consumed is generally in nanogram to picogram quantities per meal.

For translational relevance, we calculated the human-equivalent dose (HED) from our mouse model. In our study, mice weighing approximately 30 grams consumed an average of 4 grams of high-fat diet (HFD) containing 0.01% capsaicin, equating to approximately 0.04 mg of CAP per mouse per day. According to the standard dose conversion method based on body surface area normalization (referencing Nair & Jacob, J. Basic Clin Pharm. 2016;7(2):27–31), this intake corresponds to an estimated 6.4 mg of pure capsaicin (16 million SHU) per day for a 65 kg human.

Comment 5: Figures 3 and 5 are missing in the results.

Response: Thank you for bringing this to our attention. It was indeed concerning to see Figures 3 and 5 missing. Upon reviewing the submitted version, we found that the figures were intact at the time of submission. However, we have now re-uploaded the missing figures and ensured they are correctly included in the revised manuscript.

We will specifically stress this point in our cover letter and request confirmation from the journal administrators to ensure that all figures are properly included in the final submission.

Comment 6: It is suggested to expand the discussion regarding the genes and proinflammatory cytokines evaluated.

Response: The discussion regarding the evaluated genes and proinflammatory cytokines has been expanded in the revised version and is highlighted accordingly in the discussion section.

Comment 7: MASLD is also associated with high fructose diets, which also promote increased lipogenesis, dyslipidemia, and insulin resistance, so it would be interesting to discuss further the choice of diet for this study.

Response: We have included this model in the discussion and highlighted it in the revised version.
